# The effect of hearing aids on cognitive function: A systematic review

**Maxime E. Sanders**[1⊘]**, Ellen Kant**[1⊘]**, Adriana L. Smit**[1,2]**, Inge Stegeman**[1,2,3,4]*

**1** Department of Otorhinolaryngology and Head & Neck Surgery, University Medical Center Utrecht, Utrecht, The Netherlands, **2** Brain Center, University Medical Center Utrecht, Utrecht, The Netherlands, **3** Department of Ophthalmology, University Medical Center Utrecht, Utrecht, The Netherlands, **4** Epidemiology and Data Science, Amsterdam University Medical Centers, University of Amsterdam, Amsterdam, The Netherlands

⊘ These authors contributed equally to this work.

* I.Stegeman@umcutrecht.nl

## Abstract

### Rationale

Dementia currently affects 50 million people globally with this expected to triple by 2050. Even though hearing loss is associated with cognitive decline, the underlying mechanisms are not fully understood. Considering hearing loss is the largest modifiable risk factor for developing dementia, it is essential to study the effect of hearing aids on cognitive function.

### Objective

To systematically review the existing literature to examine the evidence for using hearing aids intervention as a treatment for deteriorating cognitive function.

### Design

A search of PubMed, Cochrane Library, Embase and grey literature was conducted revealing 3060 unique records between 1990–2020. Two reviewers independently selected longitudinal studies observing the effects of hearing aids on cognitive function in persons without dementia at onset of the study. Due to the heterogeneity of the data, a meta-analysis could not be performed. Outcomes are described in a summary of findings table and portrayed diagrammatically.

### Results

We identified 17 unique studies, spanning 30 years of research and 3526 participants. The included studies made use of 50 different cognitive function tests. These tests were grouped into separate cognitive domains according to the DSM-V classification for further analysis. The most beneficial impact of hearing aids seems to be in the cognitive domain of executive function, with six studies showing improvement, two studies being inconclusive and three studies not demonstrating a significant effect. Three of five studies demonstrated significant improvement when screening for brief mental status. The least beneficial impact is seen in

**Data Availability Statement:** All relevant data are within the manuscript and its Supporting information files.

**Funding:** The authors received no specific funding for this work.

**Competing interests:** The authors have declared that no competing interests exist.

domain of complex attention, with eight studies showing no significant effects, compared with one demonstrating improvement with intervention.

## Conclusions

Based on this systematic review, we conclude that there is controversy about the effects of hearing aids on cognition. Additional research through randomized clinical trials with standardized cognitive assessment and longer follow-up is warranted to further elucidate this relationship.

## Introduction

Dementia affected 50 million people worldwide in 2018, with this number expected to increase to 150 million by the year 2050 [1]. Dementia is a disorder characterized by slowly progressing cognitive decline [2] that interferes with normal daily functioning and independence [3]. The history of cognitive decline and dementia can be dated back to the ancient Greek and Roman times and was long thought to be an inevitable part of life, neither preventable nor treatable [4]. However, our understanding of cognitive decline is rapidly shifting with more insights gained into aetiology of the clinical syndrome. In 2017, the Lancet Commission for dementia prevention, intervention, and care described a model of life-course risk factors that influence the progression to dementia [3]. It is estimated that 40% of these risk factors are modifiable and could provide an opportunity for early prevention and disease progression [5]. According to the most recent report of the 2020 Lancet Commission for dementia prevention, intervention, and care, hearing loss accounts for up to 8.2% of the risk factors of dementia and is thereby the largest potentially modifiable risk factor identified [5].

Hearing loss is a public health problem affecting over 466 million people globally [6]. According to the Global Burden of Disease Study, hearing loss has become the third highest cause of years lived with disability [7]. The prevalence of hearing loss increases with older age, with an estimation by the WHO that one-third of adults older than 65 years of age experience disabling hearing loss (>40 dB at the average of 500, 1000, 2000 and 4000Hz in the better ear) [6–8]. Hearing loss is associated with reduced quality of life and can result in loneliness [9], social isolation, lowered mood, depression [10], anxiety and poorer physical health [11–13]. Conventional hearing aids are the primary treatment for hearing loss and are effective to restore hearing. Despite this, hearing aids are significantly underutilized with less than 20% of those with a hearing impairment using them [14].

It is estimated that if interventions have the ability to delay the onset of dementia by 1 year, this would decrease the global prevalence of dementia by 10% [15]. Hearing aids are a relatively low-cost intervention when compared to the high societal and psychosocial costs that accompany declining cognitive ability. Considering that hearing loss often precedes the onset of clinical dementia by at least 5 to 10 years [8], it is imperative to investigate the role of hearing intervention on cognitive outcome before the onset of dementia. Henceforth, the focus of this review will be on the stage before dementia, cognitive decline. Even though there is a strong association between hearing loss and cognitive decline [16–18], the underlying mechanisms linking the two are not fully elucidated. Previous systematic reviews have focused largely on the association between hearing loss and cognitive function and while some commented on the intervention effects of hearing aids, there was no clear consensus [5, 11, 19, 20]. The purpose of this study is to systematically review the literature on prospective cohort studies

examining the intervention effect of hearing aid usage on cognitive function in adults who have not been diagnosed with dementia.

## Methods

This review is reported according to Preferred Reporting Items for Systematic Review and Meta-Analysis [21]. The protocol for this review is pre-registered on the PROSPERO International Prospective Register of Systematic Reviews (CRD42020171872) [22].

### Eligibility criteria

Inclusion criteria included published and ongoing longitudinal studies describing the effect of hearing aid treatment on cognitive function. Studies were only included when participants were declared first-time hearing aid users. Study populations with diagnosed dementia at onset of study and duplicate data sets were excluded.

### Search strategy and information sources

The search terms "hearing aids" and "cognition" with their synonyms were used (S1 File. Syntax search string) We conducted an extensive, literature search in electronic databases (PubMed, Cochrane Library, Embase) on 11 November 2020. Grey literature was searched using two electronic databases: ClinicalTrials.gov [23] and TrialRegister.nl [24]. We supplemented the electronic search by the manual searches of the references from articles and review papers. We contacted authors of ongoing studies or articles whose full-text or data were not yet available for unpublished work.

### Study selection

The records were independently screened by two reviewers (MS and EK). Relevant articles identified in the title and abstract screening were further assessed in full-text screening and assessed for suitability according to the inclusion and exclusion criteria. Conflicts were resolved by discussion and in the case of disagreement a third researcher (IS or AS) weighed in.

### Data collection process and analysis

The following relevant data were extracted: study ID, study location, study design, number of participants, age and gender of participants, definition of the hearing loss, assessment method of hearing loss, average hearing loss at baseline, follow-up time and outcome measures concerning cognition. Due to heterogeneity of the cognitive tests used and length of follow-up, a meta-analysis could not be performed. Instead, the cognitive outcomes are described in a summary of findings table. When available, the control group data and either a probability value or confidence interval were included. We grouped all outcomes of individual cognitive tests into five (of six) neurocognitive domains according to the DMS-V classification: perceptual motor function, executive function, complex attention, language, learning and memory and two broader domains; brief mental status and general intelligence [25, 26]. Note that the social cognition domain (the sixth domain according to DSM-V) was not tested for in the studies reviewed. These data were portrayed diagrammatically. The effect was expressed as either significantly improving, impeding or having no significant effect on cognition.

### Risk of bias assessment

Two reviewers (MS and EK) independently critically appraised the studies for quality. The Cochrane Risk of Bias tool [27] was used to assess randomized controlled trials (RCTs), while the Newcastle-Ottawa Scale [28] was used for non-randomized prospective cohort studies. The Newcastle-Ottawa Scale was converted to good, fair or poor quality according to the Agency for Health Research and Quality standards. A study was considered of good quality in case of 3 or 4 stars in selection domain and 1 or 2 stars in comparability domain and 2 or 3 stars in outcome/exposure domain. A study was considered of fair quality in case of 2 stars in selection domain and 1 or 2 stars in comparability domain and 2 or 3 stars in outcome/exposure domain. A study was considered of poor quality in the case of 0 or 1 star in selection domain or 0 stars in comparability domain or 0 or 1 stars in outcome/exposure domain. Disagreements between the reviewers were resolved by discussion or input from a third researcher (IS or AS).

## Results

### Search strategy

The results of the search generated 4027 records, with an additional 7 found through grey literature searches and snowballing. Exclusion of duplicates left 3060 unique records. Title and abstract screening revealed 88 potentially relevant studies. One additional study was added through correspondence with authors. The full-text screening identified 17 relevant studies (Fig 1).

### Study design characteristics

Table 1 outlines the characteristics of the included studies. The 17 included studies span from 1990 to 2020, 4 of which are RCTs [29–32], and the other 13 are prospective cohort studies. The study sample sizes ranged from 6 to 2040 participants, totalling 3526 participants in this systematic review. The studies were situated largely in high income countries. The duration of follow-up differed from 6 weeks to 14 years. With six of the studies using a follow-up of 6

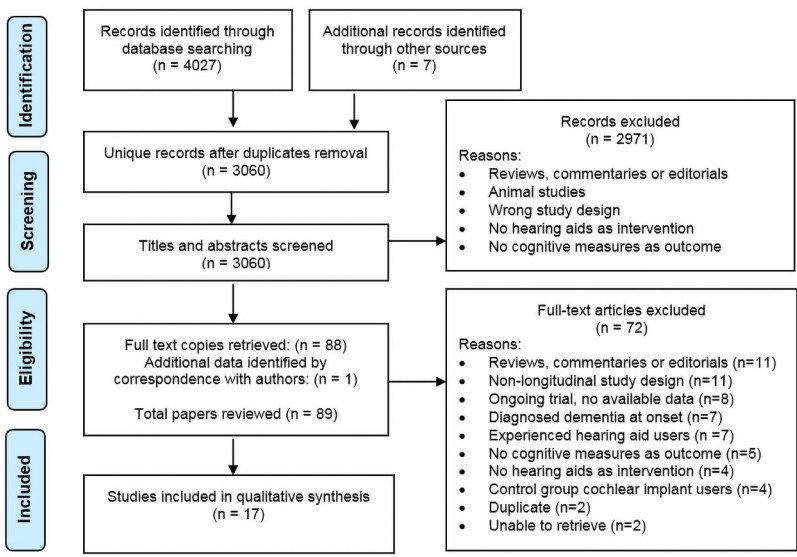

**Fig 1. Flow chart of article selection process.**

**Table 1. Study characteristics.**

| First author, year | Country | Study design | N | Age, Mean (SD) or Range, y | N Male (%) | Cognitive domains assessed | Cognitive tests | Definition hearing impairment | Hearing level (dB), Mean (SD) | Follow-up length |
|---|---|---|---|---|---|---|---|---|---|---|
| | | *Study design with control group* | | | | | | | | |
| *Mulrow et al. 1990* | USA | RCT | 194 | 73 (7) | 99 | Brief mental status | SPMSQ | ≥40 dB at 2 kHz | HA: 53 (10) | 4 months |
| | | | | | | | | | HI-C: 51 (8) | |
| *Tesch-Römer et al. 1997* | Germany | Cohort | 140 | HA: 71.8 (8.2) | HA: 42.9 | Complex attention | DSST | >30 dB at 0.5, 1, 2, or 3 kHz, *or* >30 dB in the worst ear 2x at 0.5, 1, 2, or 3 kHz | HA: 47.3 (10.4) | 6 months |
| | | | | HI-C: 71.5 (6.5) | HI-C: 46.4 | | Digit letter test | | HI-C: 37.8 (9.6) | |
| | | | | | | | Animal test | | | |
| | | | | NH-C: 69.4 (6.1) | NH-C: 47.6 | Language | Letter 's' test | | *(NB: p< 0.01)* | |
| | | | | | | | Spot-a-word test | | | |
| *Van Hooren et al. 2005* | Netherlands | Cohort | 102 | HA: 72.5 (7.3) | HA: 64.3 | Complex attention | SCWT | ≥35 dB at 1, 2, and 4 kHz | HA: 46.46 (7.30) | 12 months |
| | | | | HI-C: 74.5 (6.8) | HI-C: 63.0 | Executive function | LDST | | HI-C: 44.09 (7.69) | |
| | | | | | | | CST | | | |
| | | | | | | Learning and memory | VVLT | | | |
| | | | | | | Language | Verbal fluency test | | | |
| *Obuchi et al. 2011* | Japan | Cohort | 12 | *NI* | *NI* | Complex attention | Dichotic listening test | >40 dB | HA: 46.6 (10.8) | 3 years |
| | | | | | | General intelligence | WAIS-R | | HI-C: 31.5 (13.7) | |
| *Choi et al. 2011* | Korea | Cohort | 29 | HA: 69.5 (8.3) | *NI* | Learning and memory | Korean-VVLT: | *NI* | HA: 50.3 (14.7) | 6 months |
| | | | | HI-C: 63.1 (11.8) | | | • Total score | | HI-C: 40.7 (19.0) | |
| | | | | | | | • Recognition score | | | |
| | | | | | | | • Latency score | | | |
| *Doherty et al. 2015* | USA | Cohort | 24 | 50–74 | *NI* | Executive function | Listening span test | 2 of 3 thresholds: >26 dB at 2 kHz, >30 dB at 3 kHz and/or >35 dB at 4 kHz, and ≤15 dB difference between ears | *Displayed graphically* | 6 weeks |
| | | | | | | | N-back test | | | |
| *Dawes et al. 2015* | USA | Cohort | 666 | HA: 69.5 (9.8) | HA: 68.1 | Brief mental status | MMSE | >40 dB at 3 and 4 kHz | HA: 38.9 (10.5) | 11 years |
| | | | | | | Complex attention | DSST | | HI-C: 29.8 (9.0) | |
| | | | | Non-HA users: 68.0 (9.7) | Non-HA: 74.4 | Executive function | TMT-A | | | |
| | | | | | | | TMT-B | | *(NB: p< 0.0001)* | |
| | | | | | | Learning and Memory | AVLT | | | |
| | | | | | | Language | Verbal fluency test | | | |
| *Karawani et al. 2018* | USA | RCT | 32 | HA: 75 (6.52) | 40.6 | Executive function | NIH toolbox: | ≥25 dB from 0.5 to 4kHz; not >90 dB at any frequency | HA: 42.58 (7.15) | 6 months |
| | | | | | | | • Working memory test | | HI-C: 40.21 (73.8) | |
| | | | | | | | • Flanker test | | | |
| | | | | HI-C: 74 (5.79) | | Complex attention | • Pattern speed test | | | |

*(Continued)*

**Table 1.** (*Continued*)

| First author, year | Country | Study design | N | Age, Mean (SD) or Range, y | N Male (%) | Cognitive domains assessed | Cognitive tests | Definition hearing impairment | Hearing level (dB), Mean (SD) | Follow-up length |
|---|---|---|---|---|---|---|---|---|---|---|
| *Brewster et al. 2020* | USA | RCT | 13 | HA: 66.2 (63.1–67.5) [+] | HA: 57 | Learning and memory | RBANS: | NI | HA: 48.1 (33.3–51.9) [+] | 12 weeks |
| | | | | | | | • Delayed Memory | | | |
| | | | | | HI-C: 50 | | • Visuospatial/constructional | | HI-C: 42.5 (40.6–53.1) [+] | |
| | | | | HI-C: 78.2 (70.8–85.4) [+] | | Complex attention | • Attention | | | |
| | | | | | | | • Language | | | |
| | | | | | | Language | • Immediate Memory | | | |
| | | | | | | Executive function | Flanker: Executive function | | | |
| | *Study design without control group* | | | | | | | | | |
| *Acar et al. 2011* | Turkey | Cohort | 34 | 70.1 (4.8) | 88.2 | Brief mental status | MMSE | >40 dB at 0.5, 1, 2, 4 kHz | HA: 57.2 right, 56.3 left | 3 months |
| *Magalhães et al. 2011* | Brazil | Cohort | 50 | 60–74 & ≥ 75 | 54 | Brief mental status | MMSE | Speech Recognition Percent Index >50% | NI | 12 months |
| *Boi et al. 2012* | Italy | Cohort | 15 | 78 (4.4) | 67 | Brief mental status | MMSE | NI | HA: 14/15 pts: 58.39, 1/15 pts: range 71-90dB | 6 months |
| | | | | | | | CDT | | | |
| *Castiglione et al. 2016* | Italy | Cohort | 125 (30)* | 70–80 | (50) | Executive function | Digit span test | NI | NI | 1 month |
| *Desjardins et al. 2016* | USA | Cohort | 6 | 54–64 | NI | Executive function | Listening span test | NI | *Displayed graphically* | 6 months |
| | | | | | | | Reading span test | | | |
| | | | | | | | CRM | | | |
| | | | | | | Complex attention | ARTT | | | |
| | | | | | | | Stroop test | | | |
| | | | | | | | DSST | | | |
| *Deal et al. 2017*** | USA | RCT | 40 | 70–84 | 32.5 | Learning and memory | Delayed word recall | ≥30 dB at 0.5, 1 and 2 kHz | HA: 44 (6) | 6 months |
| | | | | | | | Incidental learning | | HI-C: 47 (10) | |
| | | | | | | Executive function | Logical memory | | | |
| | | | | | | Complex attention | TMT-B | | | |
| | | | | | | Language | TMT-A | | | |
| | | | | | | | DSST | | | |
| | | | | | | | Word fluency | | | |
| | | | | | | | Boston naming test | | | |
| *Maharani et al. 2018* | USA | Cohort | 2040 | 62.8 (7.7) | 62 | Executive function | Immediate & delayed word recall | NI | NI | 14 years |
| | | | | | | | Serial 7's | | | |
| | | | | | | | Date naming | | | |

(*Continued*)

**Table 1.** (Continued)

| First author, year | Country | Study design | N | Age, Mean (SD) or Range, y | N Male (%) | Cognitive domains assessed | Cognitive tests | Definition hearing impairment | Hearing level (dB), Mean (SD) | Follow-up length |
|---|---|---|---|---|---|---|---|---|---|---|
| *Sarant et al. 2020* | Australia | Cohort | 99 | 72.5 (4.86) | 45.5 | Executive function | Groton maze learning test | >31 dB at 0.5, 1, 2 and 4 kHz | HA: 31.24 (7.9) | 18 months |
| | | | | | | Complex attention | One back test | | | |
| | | | | | | | Identification test | | | |
| | | | | | | Learning and memory | One card learning test | | | |
| | | | | | | Perceptual motor function | Detection test | | | |

*: Castiglione et al 2016, only group A included for analysis (30 participants).

**: Comparator group is successful aging intervention.

+: Median (IQR).

HA: hearing aid users; HI-C: hearing impaired control; NH-C: normal hearing control; NI: No information.

HL: hearing loss; pt(s): participant(s).

ARTT: Auditory Reaction Time Task; AVLT: Auditory Verbal Learning Test; CDT: Clock Drawing Test; CRM: Corpus Response Measure; DSST: Digit Symbol Substitution Test; LDST: Letter Digit Substitution test; MMSE: Mini-Mental Status Examination; SCWT: Stroop coloured word test; SPMSQ: Short Portable Mental Status Questionnaire; TMT: Trail Making Test; VVLT: Visual Verbal Learning Test; WAIS-R: Wechsler Adult Intelligence Scale-Revised Short forms.

months [30, 31, 33, 36, 41, 43]. In nine studies the control group consisted of hearing-impaired participants without hearing aid treatment [29, 31–38]. The other eight studies compared the effect of the intervention results to baseline results [30, 39–45]. The studies by Deal et al. [30] and Brewster et al. [32] are primarily feasibility studies and secondarily assessed the cognitive outcomes over a period of 6 months and 12 weeks respectively. In the study of Deal et al. [30] the comparator group participated in the "successful aging intervention", an interactive health education program which did not meet the inclusion criteria for this review. Therefore, for this review, the intervention outcomes were compared with baseline outcomes for the hearing aid group. In the study of Castiglione et al. [42] only group A met the inclusion criteria for this review. The 17 included studies made use of 50 different cognitive function tests, none of which are used across the board. These outcome measures covered the following cognitive domains: brief mental status, general intelligence, perceptual motor function, executive function, complex attention, language, learning and memory.

## Assessment of risk of bias and representativeness

The risk of bias assessments for the RCTs and the prospective cohort studies are shown in Tables 2 and 3 respectively. Three RCTs are judged to have some concerns [30–32] and one as high risk for bias [29].

In all studies, participants were randomised using appropriate random sequence methods, however one trial had baseline imbalances with a significantly younger intervention group and therefore was rated with some concerns [32]. We judged the deviations from the intended intervention to be high in one study [29], with some concerns in two studies [30, 31]. In these studies blinding was not performed and in one study [29] there was insufficient assignment to the intervention, because 15% of persons in intervention group reported wearing their aids less than four hours daily. The deviations from intended intervention was judged to be low in one study [32], because of the use of sham hearing aids. We judged a low risk of bias in missing outcome data in all the RCTs [29–32], because data were available for nearly all participants.

**Table 2. Cochrane risk of bias tool for randomized trials.**

| Study | Randomization process | Deviations from intended interventions | Missing outcome data | Measurement of the outcome | Selection of the reported result | Overall |
|---|---|---|---|---|---|---|
| Mulrow et al. 1990 | + | - | + | - | +/- | - |
| Deal et al. 2017 | + | +/- | + | + | + | +/- |
| Karawani et al. 2018 | + | +/- | + | +/- | + | +/- |
| Brewster et al. 2020 | +/- | + | + | + | + | +/- |

+ Low risk +/- Some concerns—High risk.

The measurement of outcome was considered to be a high risk of bias in one study [29], because of the low sensitivity and specificity of the use of Short Portable Mental Status Questionnaire. The measurement of outcome was rated with some concern of bias in one study [31], because of potential influence on performance due to a difference in the amount of contact moments between the exposed and non-exposed group. The measurement of outcome

**Table 3. Newcastle-Ottawa Scale for assessing non-randomized trials.**

| Study ID | Selection | Comparability | Outcomes | Total | Quality |
|---|---|---|---|---|---|
| Tesch-Römer et al. 1997 | + - - ? | + - | + - + | 4 | - |
| Van Hooren et al. 2005 | + + + + | + + | + - + | 8 | + |
| Acar et al. 2011 | + - ? - | + - | + - - | 3 | - |
| Magalhães et al. 2011 | + - ? | + + | - - - | 3 | - |
| Obuchi et al. 2011 | - - ? ? | + + | + - - | 3 | - |
| Choi et al. 2011 | + + ? ? | + - | + - + | 5 | +/- |
| Boi et al. 2012 | - - + + | - - | + - + | 4 | - |
| Dawes et al. 2015 | + - - ? | + + | + + - | 5 | - |
| Doherty et al. 2015 | ? - + ? | + - | - - - | 2 | - |
| Castiglione et al. 2016 | + - ? ? | + - | + - + | 4 | - |
| Desjardins et al. 2016 | - - + + | - - | + - + | 4 | - |
| Maharani et al. 2018 | + - - + | + + | + + - | 6 | +/- |
| Sarant et al. 2020 | + - + + | + + | + - - | 6 | - |

Selection

• Representativeness of exposed cohort

• Selection of non-exposed cohort

• Ascertainment of exposure

• Outcome of exposure

Comparability

• Comparability of cohorts on the basis of the design or analysis (worth 1 or 2 stars)

Outcomes

• Assessment of outcome

• Length of follow-up

• Adequacy of follow-up of cohorts

Thresholds for converting the Newcastle-Ottawa scales to AHRQ standards (good, fair, and poor):

+ Good quality: 3 or 4 stars in selection domain AND 1 or 2 stars in comparability domain AND 2 or 3 stars in outcome/exposure domain.

+/- Fair quality: 2 stars in selection domain AND 1 or 2 stars in comparability domain AND 2 or 3 stars in outcome/exposure domain.

- Poor quality: 0 or 1 star in selection domain OR 0 stars in comparability domain OR 0 or 1 stars in outcome/exposure domain.

was judged a low risk in two studies [30, 32]. The selection of the reported result was judged low risk in three studies [30–32] and some concerns in one study [29], because in this study only one of the multiple analyses was reported numerically.

The 13 cohort studies were assessed using the Newcastle-Ottawa Scale and 1 is rated as good quality [34], 2 rated as fair quality [36, 44] and 10 rated as poor quality [33, 35, 37–43, 45]. The item selection of the non-exposed cohort was assessed with a high risk of bias in 11 of the 13 studies [35, 37–45], as these non-exposed participants were all recruited from a different source. There was insufficient data on the ascertainment of exposure in five studies [35, 36, 39, 40, 42], mainly because they did not specify how they measured adherence to hearing aid use. Additionally, there was insufficient data on outcome of exposure in seven studies [33, 35–38, 40, 42], because they did not demonstrate that cognitive decline was not present at the start of the study. On the item comparability, two studies were assessed with a high risk of bias [41, 43], five studies got rated with some concerns [33, 36, 37, 39, 42] and six studies got rated with a low risk of bias [34, 35, 38, 40, 44, 45]. The item length of follow-up had a high risk of bias, with only two studies have a follow-up longer than ten years [38, 44], which we assessed as a minimum adequate length of follow-up because of the slowly progressive nature of cognitive decline. Eleven studies were rated with a low risk of bias in the assessment of outcome [33–36, 38, 39, 41–45], in one study the assessment of outcome was performed by self-report [40] and another provided a non-blinded, auditory test, considered not sufficient [37].

## Hearing level of the participants at baseline

Table 1 describes the mean hearing levels of the participants in the studies. In most of the studies, hearing impairment is measured by pure tone audiometry. The mean hearing level in the intervention groups varied from 31.2 (7.9) dB to 58.4dB, whereas the mean hearing level in the control groups varied from 29.8 (9.0) dB to 51 (8) dB. Regarding the studies with a control group, two studies showed intervention groups with a higher reported hearing loss level than the control group (Obuchi et al. [35] HA: 46.6(10.8)dB, HI-C: 31.5(13.7)dB; Choi et al. [36] HA: 50.3(14.7)dB, HI-C: 40.7(19.0)dB). Additionally, in two other studies this was statistically tested and a statistically significant difference of about 10dB was found (Tesch-Römer et al. [33] Hearing aid group (HA) 47.3(10.4)dB, Hearing impaired control group (HI-C) 37.8(9.6) dB, p< 0.01; Dawes et al. [38] HA: 38.9(10.5)dB, HI-C: 29.8(9.0)dB, p< 0.0001). Doherty et al. [37] stated that if a participant did not meet the hearing threshold criteria for being fit with hearing aids, they were assigned to the control group. At baseline they measured the speech recognition percentage scores and stated there were no significant differences among their participant groups. In the study by van Hooren et al. [34] the hearing levels between the two groups were not matched and the control group existed of individuals who refrained from using hearing aids.

## Effect on cognition

Data of the outcomes of the cognitive tests at baseline and after intervention are provided in Table 4. Figs 2 and 3 display diagrammatically the cognitive test outcomes per study categorized in six specific cognitive domains or a generalized mental status domain. The outcomes of the studies are described per cognitive domain.

## Brief mental status domain

Five studies tested brief mental status. Mulrow et al. [29] used the Short Portable Mental Status Questionnaire and found a significant improvement in cognitive functioning after four months of follow-up (percent improvement 0.28, 95% C.I. 0.08–0.48, p = 0.008). The

**Table 4. Summary of cognitive outcomes.**

| First author, year | Cognitive tests (measuring unit*) | Hearing intervention | | | Control | | | Extra information |
|---|---|---|---|---|---|---|---|---|
| | Study design with control group | | | | | | | |
| | | Baseline | After | p-value | Baseline | After | p-value | |
| Mulrow et al. 1990 | SPMSQ | 0.47 (0.75) | 0.29 (0.66) | **p<0.05** | 0.18 (0.46) | 0.28 (0.66) | p>0.05 | Percent improvement [CI 95%], (p-value) |
| Tesch-Römer et al. 1997 | DSST | 40.1 (11.1) | 41.0 (12.1) | p>0.05 | 43.1 (12.0) | 43.6 (13.0) | p>0.05 | 0.28 [0.08–0.48] (**p = 0.008**) |
| | Digit letter test | 106.5 (23.9) | 109.3 (24.0) | p>0.05 | 110.2 (24.2) | 113.7 (26.5) | p>0.05 | |
| | Spot-a-word test | 19.5 (4.1) | 19.8 (4.3) | p>0.05 | 20.4 (3.0) | 21.2 (3.0) | p>0.05 | |
| | Letter 's' test | 19.4 (7.1) | 20.4 (8.3) | p>0.05 | 21.5 (7.8) | 21.5 (7.1) | p>0.05 | |
| | Naming animals | 26.4 (7.6) | 25.9 (7.6) | p>0.05 | 27.2 (7.3) | 28.1 (8.7) | p>0.05 | |
| Van Hooren et al. 2005 | | | | | | | | F-value (p-value): |
| | SCWT (12) | 21.81 (5.17) | 22.87 (5.78) | | 21.70 (4.61) | 21.69 (3.61) | | 5.51 (**p = 0.02** in favour of control) |
| | SCWT (I) | 30.58 (14,02) | 35.51 (24,43) | | 38.21 (25,00) | 37.88 (16,87) | | 0.15 (p = 0.70) |
| | LDST | 27.34(6.71) | 26.32 (6.82) | | 26.22 (7.87) | 24.60 (7.65) | | 0.91 (p = 0.34) |
| | CST (ab) | 28.83 (8.83) | 28.70 (9.79) | | 29.75 (8.81) | 29.91 (8.27) | | 0.09 (p = 0.76) |
| | CST (I) | 20.42 (17,40) | 22.07 (17,55) | | 17.23 (17,75) | 17.86 (19,42) | | 1.45 (p = 0,23) |
| | VVLT immediate recall | 22.89(5.97) | 25.50 (5.63) | | 23.59 (4.76) | 24.96 (5.85) | | 0.16 (p = 0.69) |
| | VVLT delayed recall | 9.64(3.19) | 10.09 (2.91) | | 8.31 (2.98) | 9.47 (3.44) | | 0.73 (p = 0.40) |
| | Verbal fluency test | 27.30(6.63) | 25.18 (6.87) | | 26.72 (6.30) | 23.22 (6.32) | | 1.54 (p = 0.22) |
| Obuchi et al. 2011 | Dichotic Listening test (% correct) | Data only presented graphically | | | | | | No significant difference between groups |
| | WAIS-R | | | | | | | Significant improvement in HA group |
| Choi et al. 2011 | K-VVLT: Total score | 32.7 (8.3) | 38.5 (11.8) | **p<0.05** | 34.4 (6.9) | 34.1 (7.3) | p>0.05 | |
| | K-VVLT: Recognition sore | 11.7 (1.9) | 13.1 (1.5) | **p<0.05** | 11.7 (2.0) | 11.3 (2.2) | p>0.05 | |
| | K-VVLT: Latency score | 7.5 (2.9) | 8.4 (2.5) | p>0.05 | 7.3 (1.9) | 8.1 (3.1) | p>0.05 | |
| Doherty et al. 2015 | Listening span test (% words recalled), N-back test | Data presented graphically | | | | | | Auditory working memory performance significantly improved with hearing aid use. |
| Dawes et al. 2015 | | | | | | | | p-value for group comparison: |
| | MMSE | 26.7 (0.4) | 25.9 (0.5) | | 26.5 (0.1) | 26.9 (0.2) | | p = 0.10 |
| | Trail-making test A (sec) | | 65.0 (7.8) | | | 57.5 (2.4) | | p = 0.37 |
| | Trail-making test B (sec) | | 147.5 (14.4) | | | 148.3 (4.4) | | p = 0.96 |
| | Auditory Verbal Learning | | 3.2 (0.5) | | | 4.1 (0.1) | | p = 0.09 |
| | DSST | | 34.0 (2.1) | | | 35.3 (0.7) | | p = 0.59 |
| | Verbal Fluency Test | | 26.2 (2.3) | | | 29.2 (0.7) | | p = 0.21 |

(*Continued*)

**Table 4.** (Continued)

| First author, year | Cognitive tests (measuring unit*) | Hearing intervention | | | Control | | | Extra information |
|---|---|---|---|---|---|---|---|---|
| Karawani et al. 2018 | Working memory | 108.10 (11.71) | 116.25 (8.99) | **p = 0.008** | 109.75 (13.01) | 107.43 (13.91) | p = 0.601 | |
| | Flanker | 109.75 (10.57) | 110.99 (13.35) | p = 0.591 | 99.82 (10.86) | 106.30 (13.07) | p = 0.151 | |
| | Processing speed | 96.90 (19.60) | 100.48 (15.23) | p = 0.308 | 88.75 (18.01) | 91.02 (19.45) | p = 0.598 | |
| Brewster et al. 2020 | RBANS: | | | | | | | Nonparametric effect size |
| | • Immediate Memory | | +19.5 (12–26)[+] | | | +16.0 (15–29)[+] | | 0.25 |
| | • Delayed Memory | | +7.5 (1–11)[+] | | | +5.5 (0–12)[+] | | 0.18 |
| | • Attention | | +4.5 (-9–9)[+] | | | +1.5 (-7–12)[+] | | 0.16 |
| | • Visuospatial/ Construction | | -6.0 (-19–-2)[+] | | | +6.0 (0–23)[+] | | 0.60 |
| | • Language | | +9.0 (-4–25)[+] | | | +7.5 (-9–16)[+] | | 0.06 |
| | Flanker: Executive function | | 0 (-1–0)[+] | | | -0.5 (-2–0) | | 0.33 |
| | Study design without control group | | | | | | | |
| | | Baseline | After | p-value | | | | |
| Acar et al. 2011 | MMSE | 20.3 (7.7) | 23.0 (7.5) | **p<0.005** | | | | |
| Magalhães et al. 2011 | MMSE total | 21.6 (3.9) | 25.3 (3.3) | **p<0.001** | | | | |
| | MMSE 60–74 years | 22.1 | 25.8 | **p<0.001** | | | | |
| | MMSE ≥75 years | 21.2 | 24.9 | **p<0.001** | | | | |
| Boi et al. 2012 | MMSE | 26.93 (0.80) | 28.17 (0.56) | NI | | | | |
| | CDT | 1.93 (0.28) | 1.93 (0.24) | NI | | | | |
| Castiglione et al. 2016* | Digit Span test | 4.80 (0.91) | 5.40 (0.89) | **p<0.005** | | | | |
| Desjardins et al. 2016 | | Data presented per participant | | | | | | N of significant improvement per test (CI 95%) |
| | Listening span test (% correct) | | | | | | | 4 / 6 |
| | Reading span test (% correct) | | | | | | | 4 / 6 |
| | Auditory reaction time (msec) | | | | | | | 2 / 6 |
| | Stroop test | | | | | | | 3 / 6 |
| | DSST | | | | | | | 4 / 6 |
| | Corpus response measure (% correct) | | | | | | | 6 / 6 |
| Deal et al. 2017 | Delayed word recall | 5.6 (1.6) | 6.1 (1.5) | Non-sig. | | | | |
| | Logical memory A | 10.5 (3.6) | 13.2 (3.8) | **p<0.001** | | | | |
| | Incidental learning | 3.5 (2.2) | 4.3 (2.2) | Non-sig. | | | | |
| | Word fluency (F, A, S) | 33.7 (12.3) | 33.6 (13.8) | Non-sig. | | | | |
| | Boston naming test | 26.7 (2.5) | 27.2 (2.5) | Non-sig. | | | | |
| | Trail making test A (sec) | 33 (29, 49.5) | 33 (27.5, 42) | Non-sig. | | | | |
| | Trail making test B (sec) | 98 (73, 109.5) | 99.5 (69.6, 118.5) | Non-sig. | | | | |
| | DSST | 40.2 (10.2) | 40.8 (11.5) | Non-sig. | | | | |

**Table 4.** (*Continued*)

| First author, year | Cognitive tests (measuring unit*) | Hearing intervention | | | Control | | | Extra information |
|---|---|---|---|---|---|---|---|---|
| *Maharani et al. 2018* | Episodic memory | Slope: memory over age: | Slope: memory over age: | | | | | Difference coefficient between slopes: 0.08, **p <0.001** |
| | | β = - 0.11 (0.00) | β = - 0.03 (0.00) | **p<0.001** | | | | Association HA-use with memory scores:<br>• β = 2.13 (0.41), **p<0.001**<br>• β = 1.53 (0.41), **p<0.001** (with RF) |
| *Sarant et al. 2020* | GML test | 58.81 (15.53) | 51 (15.35) | **p = 0.001** | | | | |
| | Detection test | 2.58 (0.08) | 2.6 (0.08) | p = 0.077 | | | | |
| | Identification test | 2.78 (0.06) | 2.78 (0.07) | p = 0.869 | | | | |
| | One card learning test | 0.94 (0.14) | 0.96 (0.11) | p = 0.262 | | | | |
| | One back test | 2.96 (0.1) | 2.94 (0.08) | p = 0.205 | | | | |

NI, No Information; Non-sig, non-significant; RF, risk factors.

* Measuring unit is stated if not same as 'points' or 'score'.

⁺ Median (IQR).

Abbreviations cognitive test: CST, Concept Shifting Task; DSST: Digit Symbol Substitution Test; GML, Groton Maze Learning; LDST, Letter Digit Substation Test; MMSE, Mini-Mental Status Questionnaire; SCWT, Stroop Coloured Word Test; SPMSQ, Short Portable Mental Status Questionnaire; (K-)VVLT, (Korean-) Visual Verbal Learning Test; WAIS-R, Wechsler Adult Intelligence Scale-Revised short form.

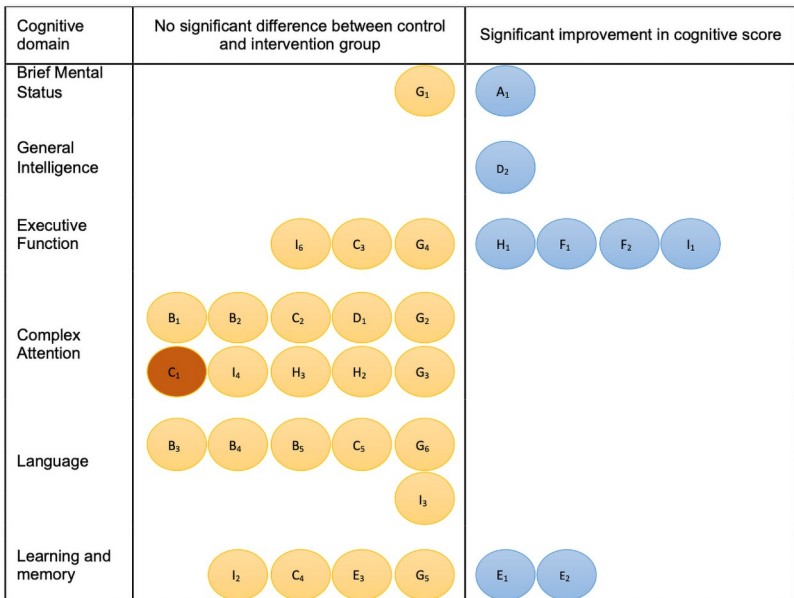

**Fig 2. Representation of cognitive tests divided into cognitive domains for studies with control group.** A: Mulrow 1990 (A₁: SPMSQ); B: Tesch-Romer 1997 (B₁: DSST, B₂: Digit letter test B₃: Spot-a-word, B₄: Letter 's' test, B₅: Naming animals); C: van Hooren 2005 (C₁*: SCWT, C₂: LDST, C₃: CST, C₄: VVLT-immediate and -delayed, C₅: Verbal fluency test); D: Obuchi 2011 (D₁: dichotic listening test, D₂: WAIS-R); E: Choi 2011 (E₁: VVLT total, E₂: Recognition score, E₃: Latency score); F: Doherty 2015 (F₁: Listening span test, F₂: N-back test);G: Dawes 2015 (G₁: MMSE, G₂: DSST, G₃: Trail making test A, G₄: Trail making test B, G₅: Auditory verbal learning, G₆: Verbal fluency test); H: Karawani 2018 (H₁: Working memory, H₂: Flanker, H₃: Processing speed); I: Brewster 2020 (I₁: RBANS-Immediate Memory, I₂: RBANS-Delayed Memory, I₃: RBANS-Language, I₄: RBANS-Attention, I₅: RBANS-Visuospatial/Constructional, I₆: Flanker Inhibitory Control and Attention Test) * C₁ showed significant improvement in favor of control group.

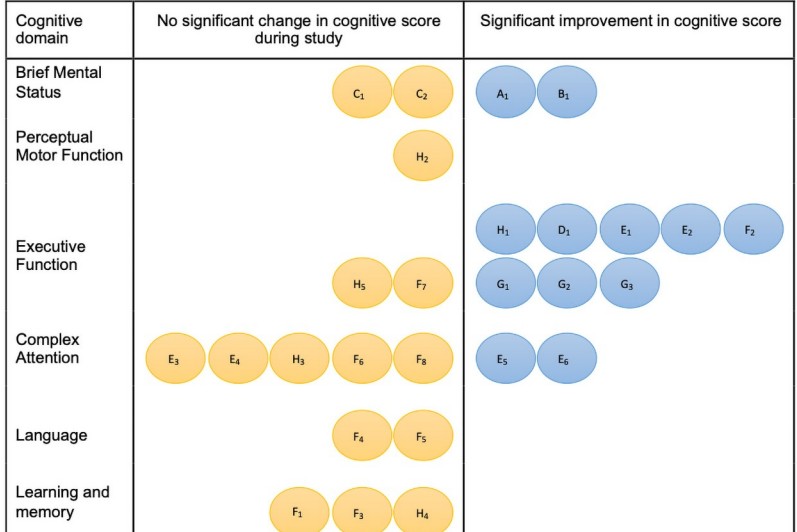

**Fig 3. Representation of cognitive tests divided into cognitive domains for studies without control group.** A: Acar 2011 (A1: MMSE); B: Magalhães 2011 (B1: MMSE); C: Boi 2012 (C1: MMSE, C2: CDT), D: Castiglione 2016 (D1: Digit Span Test); E: Desjardins 2016 (E1: Listening Span Test, E2: Reading Span Test, E3: Auditory reaction time, E4: Stroop test, E5: DSST, E6: Corpus response measure), F: Deal 2017 (F1: Delayed word recall, F2: Logical memory A, F3: Incidental learning, F4: Word fluency, F5: Boston naming test, F6: Trail making test A, F7: Trail making test B, F8: DSST); G: Maharani 2018: (G1: immediate and delayed word recall, G2: Serial 7's, G3: date naming); H: Sarant 2020 (H 1: GML test, H2: Detection test, H3: Identification test, H4: One card learning test, H5: One back test).

Mini-Mental State Examination (MMSE) was used in four studies. Acar et al. [39] and Magalhães et al. [40] both found significant improvement after respectively 3- and 12-months hearing aid use compared to baseline, respectively 20.3(7.7) to 23.0(7.5), $p<0.005$ and 21.6 (3.9) to 25.3(3.3), $p<0.001$. Furthermore Magalhães et al. [40] showed that this improvement was independent of age or gender. The study of Boi et al. [41] did not give information regarding significance, stating the score on the MMSE remained "substantially stable" after 12 months of hearing aid use compared to baseline (26.93(0.80) to 28.17(0.56)). Dawes et al. [38], with a study sample size of 666 participants, did not see any statistical difference between the intervention group and control group after the follow-up of 11 years (HA: 26.7(0.4) to 25.9(0.5), HI-C: 26.5(0.1) to 26.9(0.2)).

## General intelligence

Obuchi et al. [35] used the Wechsler Adult Intelligence Scale-Revised and found that the performance task scores of the HA user group were higher than those of the non-HA user group after three years (displayed graphically in their paper).

## Perceptual motor function

The detection test in the cohort study of Sarant et al. [45] did not reveal any significant differences after 18 months hearing aid use (2.58(0.08) to 2.6(0.08), p = 0.077).

## Executive function

Executive functioning was evaluated in five studies with a control group. Both the Concept Shifting Task [34] and the Trail Making Test-A [38] did not show a difference after respectively 12 months and 11 years hearing aid use (HA: 28.83(8.83) to 28.70(9.79), HI-C 29.75

(8.81) to 29.91(8.27), p = 0.76; HA: 65.0(7.8) HI-C: 57.5(2.4), p = 0.37). Using two auditory based tests, the listening span test and N-back test, Doherty et al. [37] showed that working memory performance significantly improved with hearing aid use after six weeks (displayed graphically in their paper). Brewster et al. [32] found a differential numerical improvement factoring active hearing aid users when testing for Immediate Memory (np-ES = 0.25), but not for the Flanker test for executive function (np-ES = 0.33). The Working Memory test used by Karawani et al. [31] significantly improved in the hearing aid group after six months (HA: 108.10(11.71) to 116.25(8.99), p = 0.008, HI-C: 109.75(13.01) to 107.43(13.91), p = 0.601).

Executive functioning was also evaluated in five studies without control groups. Maharani et al. [44], having a follow-up of 14 years, focused only on executive function, using immediate and delayed word recall tests, Serial 7's and date naming. They showed that although episodic memory declined significantly with the addition of age, the rate of decline was significantly slower after starting hearing aid use (Before hearing aid use: β = -0.11 ± 0.00, p < .001, with hearing aid use: β = -0.03 ± 0.00, p < .001, difference coefficient: 0.08, p<0.001). Hearing aid use was significantly associated with higher memory scores. (β = 2.13 ± 0.41, p<0.001). This association was independent of risk factors for cognitive impairment. Castiglione et al. [42] found a significant improvement in Digit Span test score compared to baseline (4.80(0.91) to 5.40(0.89), p<0.05) after one month hearing rehabilitation. Sarant et al. [45] used subsets of the CogState battery and found significant improvement in Groton Maze Learning test after 18 months compared to baseline (58.81(15.53) to 51(15.35), p = 0.001), but no differences in the One back test (2.96(0.1) to 2.94(0.08), p = 0.205). Deal et al. [30] found significant improvement in Logical Memory A after six months, compared to baseline (10.5(3.6) to 13.2(3.8), p<0.001) and no improvement in the Trail Making Test-B (98(73,109.5) to 99.5(69.6, 118.5). Desjardins et al. [43] used the Listening Span Test and Reading Span Test and displayed the data of their six participants individually. Five of six of the participants improved with six months of hearing aid use in both.

## Complex attention

Complex attention was assessed in six studies with control groups. After a follow-up of 12 months, van Hooren et al. [34] found that the hearing-impaired control group improved significantly better than the intervention group in the Stroop-Colour-Word Test (HA: 21.81 (5.17) to 22.87(5.78), HI-C: 21.70(4.61) to 21.69(3.61), p = 0.02). However, this difference disappeared when the analysis was conducted with the "compliant group" of participants (i.e., participants who used their hearing aid at least eight hours a day). The Letter Digit Substation Test did not show significant difference after intervention (HA: 27.43(6.71) to 26.32(6.82), HI-C 26.22(7.87) to 24.60(7.65), p = 0.34). Tesch-Römer et al. [33] showed no statistical significant differences between the intervention- and control group over a period of six months across Digit Symbol Substitution Test and Digit letter test (respectively HA: 40.1(11.1) to 41.0 (12.1), p>0.05, HI-C 43.1(120) to 43.6(13.0) p>0.05; HA: 106.5(23.9) to 109.3(24.0) p>0.05, HI-C 110.2(24.2) to 113.7(26.5), p>0.05). The Dichotic Listening test by Obuchi et al. [35] showed no significant difference (displayed graphically in their paper) after three years of follow-up. Dawes et al. [38] reported no difference between intervention and control groups both at 5- and 11-year follow-up in the Digit Symbol Substitution Test and Trail Making Test-A, and concluded that their study is "not supportive of a robust effect of hearing aid use as being protective against cognitive decline". In two tests (flanker and processing speed) by Karawani et al. [31] no differences for either group (respectively HA: 109.75(10.57) to 110.99(13.35), p = 0.591, HI-C: 99.82(10.86) to 106.30(13.07), p = 0.151; HA: 96.90(19.60) to 100.48(15.23), p = 0.308, HI-C: 88.75(18.01) to 91.02(19.45), p = 0.598). Brewster et al. [32] reported no

differential numerical improvements for the hearing aid group for the Attention test (np-ES = 0.16).

Three studies without control groups also assessed complex attention. Desjardins et al. [43], displaying the data of their six participants individually, showed improvement for 6/6 participants in Corpus response measure, 4/6 in the Digit Symbol Substitution Test, 3/6 in the Stroop test and 2/6 in the Auditory reaction time. The Digit Symbol Substitution Test and Trail Making Test-A showed no significant improvement after six months of hearing aid use in the study by Deal et al. [30] (40.2(10.2) to 40.8(11.5); 33(29,49.5) to 33(27.5,42). Complex attention measured by identification test [45] showed no improvement after 18 months hearing aid use (2.78 (0.06) to 2.78(0.07), p = 0.869).

## Language

Deal et al. [30] showed no significant improvement after six months in Word fluency and Boston naming test (33.7(12.3) to 33.6(13.8); 26.7(2.5) to 27.2(2.5)). The Verbal Fluency test, in the study van Hooren et al. [34] and Dawes et al. [38] both showed no significant improvement in the hearing intervention group after 12-months and 11-years, respectively (HA: 27.30(6.63) to 25.18(6.87), HI-C 26.72(6.30) to 23.22(6.32), p = 0.22; HA: 26.2(2.3), HI-C 29.2(0.7), p = 0.21). Tesch-Römer et al. [33] showed no difference between the intervention group and the control group six months post-intervention in the Spot-a-word, Letter 's' test, Naming animals (HA: 19.5(4.1) to 19.8(4.3), p>0.05, HI-C: 20.4(3.0) to 21.2(3.0), p>0.05; HA: 19.4(7.1) to 20.4(8.3) p>0.05, HI-C 21.5(7.8) to 21.5(7.1), p>0.05; HA: 26.4(7.6) to 25.9(7.6), p>0.05, HI-C: 27.2 (7.3) to 28.1(8.7), p>0.05) respectively). Brewster et al. [32] demonstrated no improvement after 12 weeks of hearing aids in the Language test (np-ES = 0.06).

## Learning and memory

Choi et al. [36], focusing on the cognitive domain learning and memory, found that total Visual Verbal Learning Task score and Visual Verbal Learning Task recognition score significantly improved only in the hearing aid group, respectively 32.7(8.3) to 38.5(11.8), p<0.05 and 11.7(1.9) to 13.1(1.5), p<0.05, although the VVLT latency score did not (HA: 7.5(2.9) to 8.4 (2.5), p>0.05). The Visual Verbal Learning Task -immediate and–delayed performed in the study of van Hooren et al. [34] did not show this improvement (HA: 22.89(5.97) to 25.50 (5.63), HI-C 23.59(4.76) to 24.96(5.85), p = 0.69; HA: 9.64(3.19) to 10.09(2.91), HI-C 8.31 (2.98) to 9.47(3.44), p = 0.40). Dawes et al. [38] found that the auditory verbal learning test showed no improvement after 11 years of hearing aid use as well (HA: 3.2(0.5), HI-C: 4.1(0.1), p = 0.09).

Learning and memory assessed in studies without a control group showed no significant difference in one of the tests, Delayed word recall, incidental learning [30] and one card learning test [45] (5.6(1.6) to 6.1(1.5); 3.5(2.2) to 4.3(2.2); 0.94(0.14) to 0.96(0.11), p = 0.262, respectively). Delayed Memory and the visuospatial/constructional test in the study of Brewster et al. [32] also showed no meaningful improvements (np-ES = 0 and np-ES = 0.60, respectively).

## Discussion

To study the effect of hearing aids on cognitive function in persons with hearing loss, we systematically reviewed the literature in which hearing aids were used to treat hearing loss in persons without pre-existent dementia. Seventeen studies ranging from 1990 to 2020 were included, totalling 3526 participants. Four of these were RCTs and 13 were prospective cohort studies, with the length of follow-up ranging from 6 weeks to 14 years. In this systematic review only three studies had a follow up time of longer than 12 months [35, 38, 44], namely 3,

11 and 14 years. Ten cohort studies were assessed with a high risk of bias [33, 35, 37–43, 45], and three RCTs are judged to have some concerns [30–32]. Concerning these limitations, in particular the short follow-up time, we find it premature to make definitive conclusions about the effect of hearing aids on cognitive function at this point.

In this systematic review, we provided an overview on outcomes per cognitive domain which showed that if beneficial effects of hearing aids on cognition exists, they particularly seem to affect the executive function domain. Additionally, using hearing aids might not have an effect on the language domain and little, if any, effect on complex attention and learning and memory.

Previous systematic reviews researching the relationship between hearing impairment and cognitive function diverge in their interpretations of the effects of hearing aids. Taljaard et al. 2015 [11] concluded that "at group level, individuals with treated hearing impairment demonstrate superior cognition to those with untreated hearing impairment," yet they find that the measured effect decreases when adjusting for publication bias. In contrast, in the systematic review of Mamo et al. 2018 [20] the authors concluded no significant changes in cognitive function after hearing aid use, in which conclusions were based on only three studies. Also, in the systematic review performed by Thomson et al. 2017 [19] the authors described that most of their included studies demonstrated no correlation between hearing aid use and cognition. Their study included mostly studies in which hearing aid use was determined at one time point with a yes or no question. In our study, only longitudinal studies that included participants with first-time hearing aid use were eligible. This gave more insight whether hearing aids might be preventive for cognitive decline. Our results resonate with these previous findings as we do not find a definitive answer to whether hearing aids are beneficial for cognition as a whole. However, our review does offer a new insight as we use a different perspective by not only examining the effect of hearing aids on cognition as a whole, but by examining the outcomes per cognitive domain.

It is difficult to precisely measure cognitive ability, with cognition being such a broad concept. There are many different tests available, which measure different aspects of cognition. This is reflected in the large variety of outcome measures used in the studies included in this review. There is, however, a danger to this as the selection of the tests heavily influences the results. Remarkably, many studies did not justify the reasons for the use of their tests. Furthermore, many of the tests are designed for a normal hearing population and involve either spoken instruction or involve hearing and reacting to a stimulus. The validity of these tests in a population with hearing impairment is questionable and cognitive tests relying on auditory stimuli could potentially overestimate the effects of hearing aids on cognition.

The mechanism underlying the association between hearing loss and cognitive impairment is not well understood, yet it is important to elucidate the relationship to either minimize its impact or develop preventative and rehabilitative measures. Three recent articles [2, 46, 47] describe an in-depth review of these hypotheses. These theories will only be briefly discussed in relation to the results of hearing aid treatment presented in this paper. The 'common-cause hypothesis' states that both cognitive and hearing decline are related to a common neurodegenerative process effecting both the aging brain and auditory system [2, 46]. The 'sensory-deprivation hypothesis' postulates that sensory decline causes more long-term cognitive decline. Hearing rehabilitation might reduce the risk of dementia, but the improvement or acute cessation of cognitive decline is not expected [46]. For the common cause theory, over time cognitive function should continue to decline, regardless of the use of hearing aids [2, 46]. For the majority of cognitive outcome measures, there was no significant effect of intervention. It is possible that the lack of significant benefit on the majority of cognitive domains support these hypotheses.

Conversely, the 'cascade hypothesis' states that, impoverished sensory input can lead to cognitive impairment directly or indirectly through decreased stimulation of cognitive processing [2, 47]. Restoration of hearing with hearing aids can cease the cognitive decline [2]. Additionally, the 'cognitive load hypothesis' posits that hearing loss leads to degraded auditory signals, which in turn requires greater cognitive resources for auditory processing [2, 3, 8]. This hypothesis describes a situation for hearing-impaired people which is similar to performing 'dual tasks' at the same time, resulting in cognitive reserve depletion and limiting working memory. This excessive cognitive load eventually leads to structural changes and neurodegeneration [46]. Interestingly, Griffiths et al. [47] have recently introduced an alternative hypothesis: 'the role of the medial temporal lobe'. Similar to the cognitive load hypothesis, hearing loss alters cortical activity, particularly in the medial temporal lobe. However, the critical difference is the interaction between the altered activity with dementia pathology. Use of hearing aids would not restore cognitive function in this case [47]. The information-degradation hypothesis postulates that cognitive function declines in older adults results as a consequence of compensating for diminished auditory input [46]. Difficulties in perception cascade 'upwards,' which in turn compromises higher level of cognitive processing because resources have been used for auditory perception. More effortful listening places more demand on executive function and working memory [46]. The results from our review are partly consistent with these hypotheses, showing that if a beneficial effect would exist, we would expect it to affect the specific cognitive domain of executive function.

Arguments with varying degrees of strength can be made for each hypothesis and to date none of the discussed mechanisms can be ruled out [47]. Ultimately, even though the hypotheses are presented as alternatives, it is probable that they are not mutually exclusive [48]. Johnson et al. [49] further highlights the potential importance of the auditory brain, away from the oversimplification of the dichotomy of brain and ear. The neurodegeneration of central hearing (auditory processing mechanisms) will amplify degrees of peripheral hearing loss. Potentially hearing loss is an early warning sign for dementia [49]. Considering this, we can question whether pure tone average testing is the right way to measure hearing loss in persons with cognitive decline, and whether or not hearing aids addressing peripheral hearing can address the link between hearing loss and cognitive decline. Importantly, a better understanding of the underlying mechanisms will allow for a better understanding of how realistic prevention effort may be. Furthermore, a better understanding would also lead to more accurate and consistent choices in cognitive tests, and give clearer interpretations to their results.

The overall poor quality of the included studies, in particular due to a short follow-up time, is a major limitation of our study. Bearing in mind that cognitive decline is a slowly progressing impairment, it is inadequate to draw conclusions about the effect of an intervention on cognitive function based on the presented results [3]. In this review, we included two studies with follow-up longer than 10 years; Dawes et al. [38], 11 years, and Maharani et al. [44], 14 years. The former [38] reports no significant difference between intervention and control for all outcomes measured. The latter [44] showed that although episodic memory declined with age; this rate of decline was significantly less after the start of hearing aids. It should be noted that in the study by Dawes et al. [38] there was a significant mean hearing level difference between hearing aid users and the control group (38.9dB and 29.8dB, respectively). Furthermore, the process of cognitive decline is a heterogenous, multidimensional phenomenon, and yet the large majority of the studies do not satisfactorily adjust for important confounders, such as daily activities or education level [3]. Next to this, as can be seen in the risk of bias assessments, in many studies there is no or little ascertainment of exposure. In several studies hearing aid compliance was either poorly [29, 33, 38, 44] or not described [35, 36, 39, 40, 42]. This can affect the results drastically, as the treatment group might underutilize the hearing

aids and therefore did not gain significant exposure. It is also difficult to blind either participant or researcher for the intervention, which adds another source of potential bias.

Regarding the eight studies that compared intervention to control groups, it is important to note that four studies [33, 35, 36, 38] had intervention groups with significantly worse hearing loss than the controls. Only one of these studies [36] showed a positive significant effect of intervention on cognitive outcome. Taljaard et al. [11] have described a dose-response relationship between hearing loss and cognitive function, with Golub et al. [50] finding a decrease in cognition with every 10dB reduction in hearing. Considering this knowledge, potentially more of an effect would have been seen if both control group and intervention group had similar hearing loss levels.

Dementia is a collective noun for many diseases that affect cognition. These specific diseases, such as Alzheimer's Disease, Vascular Dementia, Dementia with Lewy Bodies or Fronto-temporal Dementia, all have different aetiologies [51]. In the included articles the authors do not differentiate between the different sub-types. For the scope of this review, cognitive decline is defined as a heterogeneous clinical syndrome. For the future it would be interesting to evaluate the different subtypes of dementia and the link to their underlying pathophysiology's.

Considering hearing loss is considered to be one of the largest modifiable risk factors for developing dementia [5, 6], it is essential to study the effect of intervention on populations before they reach the stage of dementia. One of the strengths of our review is strict inclusion criteria, in which only participants without pre-existent dementia were included. This allows us to review the intervention effects of hearing aids on cognitive decline. Substantial heterogeneity remained, however. Furthermore, due to the large variety of cognitive tests and lengths of follow-up, the value in purely comparing the included studies is limited. It was for this reason that we divided the cognitive outcome measures into the seven broad cognitive domains to permit comparison. Not only does this give insight into potential domains involved in association of hearing loss, cognitive function and the effect of hearing aids, but can also provide the foundation for future studies considering which cognitive test measures to include.

## Clinical applications and further research

A key message from the Lancet reports [3, 5] is to be ambitious in terms of prevention, after all 'prevention is better than cure.' Considering that both hearing loss and dementia are separately big public health problems, it would be folly not to treat for hearing loss. Additionally, depression, social isolation and physical inactivity are considered separate modifiable risk factors for dementia [3, 5] and seeing as hearing loss also affects physical and mental health [7, 9–12], treatment with hearing aids could have beneficial effects in multiple dimensions. Fundamentally, hearing aids are primarily used to benefit communication and social functioning. Additionally, hearing aids are known to positively affect mood and quality of life [39, 52]. Despite this, close to four-fifths of persons with hearing loss do not use hearing aids [14]. These low rates might be caused by the stigmatization, underestimation of the benefits of hearing rehabilitation and the lack of access [53]. Seeing as hearing loss is easily diagnosable and treatable, we advise clinicians to be vigilant for hearing loss and to counsel patients on the benefits of hearing aids.

Regarding future research, firstly, there is a need for long-term, large-scale observational studies to better understand the effects of hearing aids. Cognitive decline can only be adequately observed in studies that span multiple years, if not decades [48]. Ideally, this would be researched in the form of a randomized clinical trial. On this note, the ACHIEVE trial is currently ongoing, a trial of 850 older adults randomized to either a hearing intervention or

successful aging intervention, being an interactive health education program, to study the effects on cognition [30, 54]. Secondly, as is clear from the wide variety of cognitive measures discussed in this review, there is a need for a standardization of well-justified cognitive measures to use for this purpose. Thirdly, further elucidation of the underlying mechanisms is necessary, and an important step for further research is the stratification of results according to subtypes of dementia. Fourthly, even though domains of social and emotional cognition have not been focused on in wider literature, it would be relevant to evaluate the effects of hearing aids on these as well. Finally, as the 2020 Lancet Commission on dementia prevention, intervention, and care recognizes hearing loss to be a mid-life risk factor for dementia and considering hearing loss starts around mid-life, it would be interesting to conduct studies in slightly younger adults [5].

## Conclusion

The included studies in this systematic review had a high risk of bias, in particular due to a short time of follow up. Therefore, definitive conclusions about the effects of hearing aids on cognition or the prevention of cognitive decline are difficult to draw. The precise effect of hearing aids on cognitive function is debatable.

## Supporting information

**S1 File. Syntax search string.**
(DOCX)

**S2 File. PROSPERO international prospective register of systematic reviews.**
(PDF)

**S1 Checklist. PRISMA checklist.**
(DOC)

## Author Contributions

**Conceptualization:** Maxime E. Sanders, Ellen Kant, Adriana L. Smit, Inge Stegeman.

**Data curation:** Maxime E. Sanders, Ellen Kant.

**Formal analysis:** Maxime E. Sanders, Ellen Kant.

**Investigation:** Maxime E. Sanders, Ellen Kant.

**Project administration:** Maxime E. Sanders, Ellen Kant.

**Supervision:** Maxime E. Sanders, Ellen Kant, Adriana L. Smit, Inge Stegeman.

**Writing – original draft:** Maxime E. Sanders, Ellen Kant.

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
