## [Decision Letter · Decision Letter 0]

10 Jun 2021

PONE-D-21-13908

The effect of hearing aids on cognitive function: a systematic review

PLOS ONE

Dear Dr. Stegeman,

Thank you for submitting your interesting manuscript to PLOS ONE. The reviewers have raised a number of issues, as you will see below. After careful consideration, therefore, we feel that it has merit but does not fully meet PLOS ONE’s publication criteria as it currently stands. Therefore, we invite you to submit a revised version of the manuscript that addresses the points raised during the review process.

We look forward to receiving your revised manuscript.

Kind regards,

Antony Bayer

Academic Editor

PLOS ONE

Journal Requirements:

 [The author(s) received no specific funding for this work.].

Additional Editor Comments (if provided):

Reviewers' comments:

Reviewer's Responses to Questions

**Comments to the Author**

1. Is the manuscript technically sound, and do the data support the conclusions?

Reviewer #1: Yes

Reviewer #2: Partly

Reviewer #3: Yes

Reviewer #4: Yes

2. Has the statistical analysis been performed appropriately and rigorously? 

Reviewer #1: N/A

Reviewer #2: N/A

Reviewer #3: Yes

Reviewer #4: Yes

3. Have the authors made all data underlying the findings in their manuscript fully available?

Reviewer #1: Yes

Reviewer #2: No

Reviewer #3: Yes

Reviewer #4: Yes

4. Is the manuscript presented in an intelligible fashion and written in standard English?

Reviewer #1: Yes

Reviewer #2: Yes

Reviewer #3: Yes

Reviewer #4: Yes

5. Review Comments to the Author

Reviewer #1: This is a well written and interesting manuscript on an increasingly important topic. I have only minor comments below. I would be happy to review this paper again.

48 - described a model of life-course risk factors.

50 - Hearing loss is estimated to account for up to 8.2% of dementia prevalence.

Suggest making it clear that 8.2% come from 2020 update of Lancet Commission. Also, not clear why citation #6 is at end of sentence on line 51?

54 – third highest cause?

56 – suggest also including frequencies.

386 – should this be ‘describe an in-depth’.

388 – common cause hypothesis refers to a common degenerative process effecting both the ear and brain.

390 – ‘cognitive decline’.

391 – In reference to the sensory deprivation hypothesis, I would suggest avoiding the conclusion that cognitive decline would continue regardless of hearing aid use. This hypothesis refers to a more permanent or long/chronic change in comparison to the information-degradation hypothesis, but not necessarily completely permanent. Wayne et al (2015) is referenced here but also see Lindenberger et al (1994) and Humes et al (2013) (both referenced in Wayne).

415 – although.

60/449 – suggest being consistent within the manuscript with the statistics referenced – i.e. use one, or both in both sentences.

Reviewer #2: Thank you for the opportunity to review this manuscript. The topic is relevant to clinicians and important in ongoing efforts to improve outcomes for people with dementia. Hearing loss is very common in this population, and mostly unaddressed.

The paper has evaluated a range of studies meeting inclusion criteria. The review has been registered in PROSPERO and follows acceptable reporting guidelines. Risk of bias/quality assessment of included studies was undertaken.

The authors may want to consider the following comments to improve their manuscript.

Introduction:

Several reviews have already been done on this topic – how is this review different? The authors need to clearly spell out what new information their review offers. It is not obvious.

This focusses on the clinical syndrome but makes no mention of the heterogeneity of the underlying disease pathology that may impact on the impact of HA on cognition. The range of dementia sub-types should be addressed, with specific reference to those which may impact on central auditory pathways.

The introduction focuses on dementia, yet the review is about the impact of HA on cognition in adults without dementia. Hence, the introduction should downplay the dementia emphasis, or highlight ‘dementia prevention’ from the outset, including the lack of disease modifying therapies, rather than focusing on prevalence issues.

It should be stated at the outset that this was only a narrative review and that a meta-analysis was not performed. This should be mentioned in the abstract and title as well to indicate the strength of the findings.

General:

A general concern is that the research question addresses the impact of HA use on cognition in any adult without dementia. The question is flawed. Instead, the question of interest is the impact of HA use on cognition or risk of dementia in people at risk of dementia. These group potentially follows a different cognitive decline trajectory than those not at risk, thus, the impact (and likely, the mechanism of action of HA) may differ. This needs to be discussed as a limitation.

Minor comments: (I have pointed out only a few of the several language errors; the authors should ideally review the full manuscript again for such errors).

The term ‘massively underutilized’ is slightly colloquial - consider ‘significantly underutilized’ instead.

This sentence is incomplete: ‘Especially considering that hearing aids are a relatively low-cost intervention when compared to the high 70 societal and psychosocial costs that accompany declining cognitive ability’…??

Line 98 and 106, should read ‘the date WERE extracted’ (data is plural). [Check the full manuscript for this error]

Line 140 –‘Only group A 141 from the study of Castiglione et al.[42] met the inclusion criteria for this review’ – does this meant that the same procedure was followed as the Deal study? Please be explicit about this.

Line 152– error ‘therefor’

Line 153 – ‘We judged the deviations from THE intended intervention to be high..’

Line 154 ‘there was insufficient assignment to the intervention’ …please clarify what this means - you mean the groups were imbalanced?

Line 158 - write out ‘SPQM’ in full please

Line 161 ‘ low risk in three STUDIES [30-32] and some concerns in one STUDY [29],

Line 460 ‘ Lancet Commissions on Dementia’ please use the correct title for this paper and reference appropriately.

Reviewer #3: This paper systematically evaluates and synthesises data from extant studies assessing the effects of hearing aid use on overall cognitive ability and specific domains of cognition. The paper addresses a major public health and clinical controversy and will be a timely and useful resource foe the field. Although it does not resolve the fundamental question whether hearing aid use can slow cognitive decline, the synthesis of Sanders et al. illustrates that the issues around the question are much more complex and nuanced than is often supposed and signposts directions that future research will need to take in order to resolve these issues. The focus on populations before onset of clinical dementia and the examination of specific cognitive domains are particular strengths of the study. I have a few comments intended to further strengthen the paper and contextualise it a little more clearly.

1. With respect to the information presented about the reviewed studies, it would be relevant to highlight hearing aid compliance wherever this was recorded – and if not recorded, that should be flagged as a limitation of the study. Hearing aids even when prescribed are frequently under-utilised and it cannot be assumed that the ‘treatment’ group actually was exposed to the treatment (or what ‘dose’ they actually received) without more detailed information.

2. The authors might finesse a little more their prior hypotheses about the likely effects of hearing aid use on particular cognitive domains, and discuss the findings accordingly. For example, a priori it is not particularly surprising that episodic memory is not much affected by hearing aids. On the other hand, if hearing aid use has no impact on the decline of language skills that might be telling us something quite important.

3. Although cognitive function (and more particularly, cognitive test performance) is the main focus of this review it might be worth indicating explicitly that this is not, of course, the whole story in assessing the benefit of hearing aids – subjective measures of wellbeing and mood would also need to be taken into account as well as more specific impacts on interpersonal communication and relationships and social behaviour. It might also be worth mentioning that there are other domains of cognitive function – such as social and emotional cognition – that may also be relevant but as yet have hardly been studied in relation to hearing aid use in older people.

4. In considering why the results are so mixed, the authors should consider a further explanation: that hearing aids do not address aspects of central hearing (auditory brain) function that may also be highly relevant to the link between hearing impairment and cognition. In fact, to the extent that central auditory factors interact with peripheral hearing loss, an inconsistent magnitude of benefit from peripheral amplification is exactly the pattern one might expect. This will only be resolved by:

i) the availability of adequate tests of central hearing and interventions (such as ‘smart’ hearing aids) that address central hearing, and their implementation in large scale trials of people at risk of dementia - at present, it is not clear which tests and/or devices will be optimal for this purpose though it is interesting to note that ‘smart’ hearing devices would be much easier to ‘blind’ (for the purpose of RCTs) than hearing aid use per se

ii) much more careful stratification of hearing function by proximity to the clinical onset of dementia – as it is possible that one reason hearing aids do not consistently ‘prevent’ dementia is that hearing loss is itself an early warning sign of dementia (i.e., auditory brain dysfunction)

I would recommend the authors broaden their Discussion to incorporate these considerations – as they rightly point out, the various putative mechanisms for linking hearing impairment to cognitive decline are by no means mutually exclusive.

In this regard, there are two recent relevant position papers presenting fresh hypotheses about the linkage that should be cited and discussed:

i) Griffiths TD, Lad M, Kumar S, Holmes E, McMurray B, Maguire EA, Billig AJ, Sedley W. How can hearing loss cause dementia? Neuron 2020; 108(3): 401-412. doi: 10.1016/j.neuron.2020.08.003.

ii) Johnson JCS, Marshall CR, Weil RS, Bamiou DE, Hardy CJD, Warren JD. Hearing and dementia: from ears to brain. Brain 2021; 144(2): 391-401. doi: 10.1093/brain/awaa429.

Reviewer #4: Authors have systematically reviewed the existing literature to investigate the evidence for using hearing

aids intervention as a treatment for further delaying or arresting further cognitive decline. Well-designed study. Appropriate tools have been used for the data analysis. A thorough assessment has been conducted on risk and bias of the studies. Very well-written manuscript. Authors have provided good directions for future research.

6. PLOS authors have the option to publish the peer review history of their article (what does this mean?). If published, this will include your full peer review and any attached files.

Reviewer #1: No

Reviewer #2: No

Reviewer #3: No

Reviewer #4: No

---

## [Author Response · Author response to Decision Letter 0]

26 Jul 2021

Dear Editor,

We would like to thank the editor for the opportunity to resubmit the manuscript entitled ‘The effect of hearing aids on cognitive function: a systematic review’. 

We received valuable comments of the reviewers of PlosOne and revised the paper carefully according to these comments to improve the quality of this manuscript. We would like to offer this improved manuscript to PlosOne for reconsideration of publication. Our detailed responses to the reviewers’ comments of the previous submission can be found in the document 'Response to Reviewers'. 

Thank you for considering our manuscript for publication. 

Sincerely,

On behalf of the author-team,

I. Stegeman

---

## [Editor Report · Decision Letter 1]

16 Nov 2021

PONE-D-21-13908R1The effect of hearing aids on cognitive function: a systematic reviewPLOS ONE

Dear Dr. Stegeman,

Thank you for submitting your manuscript to PLOS ONE and for your careful attention to the reviewers' comments. There are some minor issues with your revised version, specifically:line 117-119 DSM-V does not include general intelligence but does include social cognition as a cognitive domain. Please rephrase. (Similarly, on line 156-8, if the papers did not include social cognition then not all cognitive domains were covered).line 119. THESE data were portrayed...line 172. The sentence started "We judged the measurement..." does not seem to make sense.line 177. was REPORTED numerically.line 424. medial temporal LOBEline 440. spell out PTAline 462. ...UNDERUTILIZE the hearing aids and THEREFORE...line 477. defined AS a HETEROGENEOUS clinical syndrome After careful consideration, therefore, we feel that your manuscript has merit but does not fully meet PLOS ONE’s publication criteria as it currently stands. Therefore, we invite you to submit a revised version of the manuscript that addresses the points raised during the review process.

We look forward to receiving your revised manuscript.

Kind regards,

Antony Bayer

Academic Editor

PLOS ONE
---

## [Author Response · Author response to Decision Letter 1]

25 Nov 2021

Dear Editor,

We would like to thank you for the chance to resubmit the manuscript ‘The effect of hearing aids on cognitive function: a systematic review’. Our response to the review comments can be find below. 

Thank you for considering our manuscript for publication. 

Sincerely,

On behalf of the author-team,

I. Stegeman

 

Review comment Response

line 117-119 DSM-V does not include general intelligence but does include social cognition as a cognitive domain. Please rephrase. (Similarly, on line 156-8, if the papers did not include social cognition then not all cognitive domains were covered).

 Thank you for bringing this to our attention. Rectified. Line: 109-113 and 151.

line 119. THESE data were portrayed...

 We changed the text accordingly. Line: 113

line 172. The sentence started "We judged the measurement..." does not seem to make sense.

 Thank you for hightlighting this unclarity. The text has been refrased. Line: 166-171

line 177. was REPORTED numerically.

 We changed the text accordingly. Line: 173

line 424. medial temporal LOBE

 We changed the text accordingly. Line: 418

line 440. spell out PTA

 Has been adjusted to “Pure tone average” Line: 433

line 462. ...UNDERUTILIZE the hearing aids and THEREFORE...

 We changed the text accordingly. Line: 453

line 477. defined AS a HETEROGENEOUS clinical syndrome

 We changed the text accordingly. Line: 467

---

## [Editor Report · Decision Letter 2]

29 Nov 2021

The effect of hearing aids on cognitive function: a systematic review

PONE-D-21-13908R2

Dear Dr. Stegeman,

We’re pleased to inform you that your manuscript has been judged scientifically suitable for publication and will be formally accepted for publication once it meets all outstanding technical requirements.

Kind regards,

Antony Bayer

Academic Editor

PLOS ONE
---

## [Editor Report · Acceptance letter]

20 Dec 2021

PONE-D-21-13908R2 

The effect of hearing aids on cognitive function: a systematic review 

Dear Dr. Stegeman:

I'm pleased to inform you that your manuscript has been deemed suitable for publication in PLOS ONE. Congratulations! Your manuscript is now with our production department. 

Kind regards, 

on behalf of

Professor Antony Bayer 

Academic Editor

PLOS ONE